# Piezoelectricity of Bi_2_Se_3_ Nanosheet

**DOI:** 10.3390/nano13182504

**Published:** 2023-09-05

**Authors:** Tingting Jia, Liu Yang, Juncheng Zhang, Hideo Kimura, Hongyang Zhao, Quansheng Guo, Zhenxiang Cheng

**Affiliations:** 1School of Materials Science and Engineering, Hubei University, Wuhan 430062, China; jia.tingting@hotmail.com; 2Shenzhen Institutes of Advanced Technology, Chinese Academy of Sciences, Shenzhen 518055, China; 3Optics and Optoelectronics Laboratory, Department of Physics, Ocean University of China, Qingdao 266100, China; 4School of Environmental and Material Engineering, Yantai University, Yantai 264005, China; 5Hubei Key Laboratory of Plasma Chemistry and Advanced Materials, Department of Materials Science and Engineering, Wuhan Institute of Technology, Wuhan 430205, China; 6Institute for Superconducting & Electronic Materials, University of Wollongong, Innovation Campus, Wollongong, NSW 2500, Australia

**Keywords:** bismuth selenide, piezoelectricity, piezoresponse force microscopy, mechanical exfoliation

## Abstract

Bi_2_Se_3_, one of the most extensively studied topological insulators, has received significant attention, and abundant research has been dedicated to exploring its surface electronic properties. However, little attention has been given to its piezoelectric properties. Herein, we investigate the piezoelectric response in a five-layer Bi_2_Se_3_ nanosheet using scanning probe microscopy (SPM) techniques. The piezoelectricity of Bi_2_Se_3_ is characterized using both conventional piezoresponse force microscopy (PFM) and a sequential excitation scanning probe microscopy (SE-SPM) technique. To confirm the linear piezoelectricity of Bi_2_Se_3_ two-dimensional materials, measurements of point-wise linear and quadratic electromechanical responses are carried out. Furthermore, the presence of polarization and relaxation is confirmed through hysteresis loops. As expected, the Bi_2_Se_3_ nanosheet exhibits an electromechanical solid response. Due to the inevitable loss of translational symmetry at the crystal edge, the lattice of the odd-layer Bi_2_Se_3_ nanosheet is noncentrosymmetric, indicating its potential for linear piezoelectricity. This research holds promise for nanoelectromechanical systems (NEMS) applications and future nanogenerators.

## 1. Introduction

Piezoelectricity, the linear coupling of mechanical deformation and electric fields first discovered in 1957 [1,2], has become significantly important in the growing demand for high-performance and miniaturized devices such as nanoelectromechanical systems (NEMS) [3,4,5] and electronics. Two-dimensional (2D) materials with noncentrosymmetry, particularly transition-metal dichalcogenides (TMDCs), have emerged as ideal candidates for low-dimensional piezoelectric materials [6]. Piezoelectricity in two-dimensional materials has led to investigations into novel low-dimensional systems like nanotubes and single molecules [7,8]. For example, free-standing monolayer MoS_2_ membranes exhibit molecular piezoelectricity only in odd-layer configurations where inversion symmetry breaking occurs [9]: WSe_2_ possesses noncentrosymmetry and potential for linear piezoelectricity [10]. However, the limitation of symmetry in bulk materials often prevents their piezoelectric behavior, whereas atomic-level thin-layer structures can break symmetry and exhibit anomalous piezoelectricity [11]. First-principle calculations have shown that the 2D Janus Te_2_Se monolayer and its multilayers exhibit a large in-plane and out-of-plane piezoelectric response due to structural symmetry breaking and flexible mechanical properties [11]. Moreover, density functional theory (DFT) calculations have predicted the potential for a ferroelectric state and piezoelectric response in bulk Bi_2_O_2_Se under an in-plane biaxial strain, which has been experimentally supported by ultrathin free-standing nanosheets of Bi_2_O_2_Se breaking local inversion symmetry [12]. While the research on these new materials has expanded the range of piezoelectric materials, little attention has been given to exploring the piezoelectricity of thinner layers of Bi_2_Se_3_, a topological insulator (TI) known for its topological surface states and large bandgap (0.3 eV) in bulk [13,14]. Thus, the aim of this work is to investigate and provide evidence for the piezoelectric properties of Bi_2_Se_3_ nanosheets.

Piezoresponse force microscopy (PFM) is utilized to characterize piezoelectric properties in low-dimensional nanomaterials by measuring the sensitivity between the sample and tip [15]. However, since it works in contact mode, the contact resonance is significantly influenced by both material heterogeneity and surface topography [16,17,18]. PFM operates by examining the dynamics of its cantilever, influenced by the interaction between the sample and the tip. These interactions can be accurately described using the Simple Harmonic Oscillator (SHO) model [19]:(1a)$$A\left(\omega \right)=\frac{A_{0}\omega_{0}^{2}}{\sqrt{{\left(\omega_{0}^{2}-\omega^{2}\right)}^{2}+{\left(\frac{\omega_{0}\omega }{Q}\right)}^{2}}},\;$$
(1b)$$\emptyset \left(\omega \right)=\tan^{-1}\left[\frac{\omega_{0}\omega }{Q\left(\omega_{0}^{2}-\omega^{2}\right)}\right]+\emptyset_{0},$$
where $A_{0}$, $\emptyset_{0}$, *Q*, and $\omega_{0}$ represent the intrinsic amplitude, phase, quality factor, and resonance frequency of the system, respectively. By utilizing the above equations, the amplitude $A\left(\omega \right)$ and phase $\emptyset \left(\omega \right)$ can be determined at excitation frequency  ω. To overcome resonance shift during scanning, the Dual Amplitude Resonance Tracking (DART) model measures frequencies $\omega_{1}$ and $\omega_{2}$ on both sides of the resonance. The difference in the amplitudes, $A_{1}-A_{2}$, is then utilized for feedback control, which is maintained at zero in our implementation [19]. However, when encountering significant topography variation in materials, artifacts may still be observed despite conducting scans at a slow pace and carefully adjusting parameters to ensure reliable tracking.

With advancements in technology, a novel technique called sequential-excitation piezoresponse force microscopy (SE-PFM) has been recently proposed, modifying the DART model for the characterization of piezoelectric properties in fine structures [20,21,22]. Therefore, this work aims to address three aspects of this problem. First, we employ an optical microscope (OM), Raman spectra, atomic force microscopy (AFM), and a transmission electron microscope (TEM) to identify the morphology and thickness of the Bi_2_Se_3_ nanosheet. Secondly, we utilize the dual amplitude resonance tracking (DART) [19,23] mode and sequential excitation (SE) [20,21,24] modes, along with switching spectroscopy piezoresponse force microscopy (SS-PFM) and point-wise measurements [20] to characterize the piezoelectricity. Thirdly, we perform a relaxation experiment to confirm that the electromechanical response in Bi_2_Se_3_ stems from intrinsic piezoelectricity. Finally, based on the research results, we explain piezoelectricity through first-principle calculations. Although the bulk material possesses a center-symmetric structure, it loses its center symmetry due to structural distortion when the number of layers is reduced to a certain level. For instance, the existence of a surface termination [25] within a single quintuple layer (QL) or surface doping [26] on top of materials induces a piezoelectric effect. This paper significantly expands the research scope of topological insulator materials and their effects. highlighting the promising prospects of topological insulators in nanoelectromechanical systems (MEMS) in the future.

## 2. Materials and Methods

### 2.1. Sample Preparation

The fresh Bi_2_Se_3_ bulk was purchased from Shenzhen six-carbon technology Co., Ltd. (Shenzhen, China). The Si/SiO_2_ substrate (n-type 300 nm thick SiO_2_) was purchased from Suzhou silicon electronics technology Co., Ltd. (Suzhou, China). The 3M Scotch tape (38 mm × 10 m) was purchased from 3M Hong Kong Limited (Hong Kong, China). The first step involved cleaning the silicon wafer: The silicon wafer was cut into 5 mm × 5 mm sizes using a diamond knife. Subsequently, it was subjected to ultrasonic cleaning in a sequence of acetone solution (chemical purity), ethanol solution (chemical purity), and deionized water for 15 min each to remove contaminants from the silicon surface. The silicon wafer was then dried using an oil-free air compressor and placed into a plasma cleaning machine (TS-PL02) for 1 min to bombard the silicon surface with plasma, enhancing cleanliness. The second step was micromechanics stripping: The sample material was obtained by tearing and folding the bulk material with Scotch tape. The plasma-cleaned silicon wafer was placed onto the unfolded tape, and gentle pressure was applied a few times to remove the air bubbles and ensure tight contact between the tape and the substrate. A tape adhesive with Bi_2_Se_3_ is placed on a heating table (C-MAGHS7) of 55 °C for 1 min to allow contact with the substrate. After five minutes, the tape was gently peeled off, leaving behind the desired materials. The last step involved treating the sample: The peeled materials and silicon wafer were soaked in acetone and heated on a heating table at 55 °C for 20 min. This process was repeated with fresh acetone and deionized water to ensure thorough cleaning of the sample. Any remaining type on the surface of the Bi_2_Se_3_ was removed by placing the cleaned Bi_2_Se_3_ in a rapid annealing furnace and annealing it at 300 °C for 1 h under argon protection.

### 2.2. Structural Characterization

The morphology and structure properties of Bi_2_Se_3_ nanosheets were characterized using optical microscope (OM), Raman spectroscopy, and atomic force microscopy (AFM, Asylum Research, Cypher-ES, Santa Barbara, CA, USA). For OM, the sample was observed using a 60× objective lens and a 10× eyepiece. Raman spectra were collected using a Horiba Jobin Yvon LabRAM HR800 spectrometer with a He-Ne laser (632.8 nm) at room temperature. AFM was performed in a contact mode to obtain topographic images. A scanning area of 5 × 5 μm^2^ was covered using an Asyelec.01-R2 cantilever probe with a spring constant of 2 N/m and a resonance frequency of approximately 74 kHz in air.

### 2.3. Piezoresponse Force Microscopy

#### 2.3.1. Dual Amplitude Resonance Tracking Piezoresponse Force Microscopy (DART-PFM)

The DART-PFM measurements were performed using a Cypher AFM. The probe tip is coated with Ti/Ir to ensure conductivity. During the DART-PFM measurements, an AC bias with an excitation amplitude of 2 V was applied at near-resonant frequency of the probe. This AC bias excites the sample, and the resulting displacement of the probe was recorded at the same frequency. The contact resonance frequency was tracked using the dual-AC resonance tracking technique. The piezoresponse of the sample was enhanced and amplified by the quality factor. The dynamics of the cantilever motion were characterized using built-in lock-in amplifiers, which transformed the full-time-domain information into limited frequency-domain data in terms of raw amplitudes and phases. The SHO model was then used to calculate the intrinsic amplitude, phase, resonant frequency, and quality factor. Both vertical and lateral PFM modes were employed in this experiment. Representative DART mappings can be found in Appendix A.

#### 2.3.2. Sequential Excitation Scanning Probe Microscopy (SE-SPM)

The SE-PFM measurement was performed using a Cypher AFM with a 2 V AC voltage applied to the cantilever (Asyelec.01-R2) under the vertical mode. A predefined amplitude of AC voltage with a range of sequential frequencies was generated by an Arbitrary Waveform Generator (AWG) as an excitation onto the cantilever. A series of mapping was then recorded. Based on the mapping of resonant frequency obtained from the DART-PFM measurement, it was observed that the resonant frequency in the targeted area was concentrated around 340 kHz. Therefore, a range of frequencies from 325 to 355 kHz was used as the excitation frequencies for the SE-PFM measurement. 

#### 2.3.3. Point-Wise Measurements

The first and second harmonic PFM measurements were carried out using the Cypher AFM. These measurements involve measuring the frequency-dependent amplitude and phase response of probe deflection. In the first harmonic test, a linear response is observed, indicating that the tip deflection and excitation frequencies are the same. The excitation frequencies are swept around the tip-sample resonance frequency. In the second harmonic measurement, the excitation frequencies are half of the detection frequencies. The amplitude and phase transfer functions are fitted to a SHO model. For the vertical measurement, an excitation bias range of ~2.3 V is used, and the measurements are carried out at six spatial points across the Bi_2_Se_3_ nanosheets and the SiO_2_ substrate. For the lateral measurement, an excitation bias range of approximately 4 V is used, and the measurements are performed at ten spatial points.

#### 2.3.4. Switching Spectroscopy Piezoresponse Force Microscopy (SS-PFM)

For SS-PFM measurements, a sequence of DC voltages in the form of a triangle sawtooth wave was applied to the sample surface on top of the AC voltage. This was carried out using a harder Asyelec.01-R2 conductive probe to reduce the crosstalk. During the “Off state” when the DC voltage was stepped back to zero [4], the phase and amplitude response of the sample surface were measured. Comparing the measurements in the “Off state” to those in the “On state”, a noticeable change is observed. This change serves as clear evidence of the minimization of electrostatic effects.

#### 2.3.5. DFT Calculations

First-principle calculations were performed in the framework of the density-functional theory as implemented in the Vienna ab initio simulation package (VASP). The calculations employed the generalized gradient approximation (GGA) formulated by Perdew, Burke, and Ernzerhof (PBE). Projector augmented wave (PAW) potentials were used, with a kinetic cutoff energy of 400 eV. The first Brillouin zone was sampled using a 7 × 7 × 1 Monhkorst-Pack scheme. The atomic structures were relaxed using the recommended conjugate-gradient algorithm until the maximal atomic force component acting on each atom is less than 0.01 eV/Å. To minimize interferences from neighboring layers, the thickness of the vacuum layer along the *z*-axis was set to be larger than 15 Å. The elastic tensor coefficients, *C_ij_*, were calculated using the finite difference method, and the strain coefficients of the piezoelectric tensor, *e_ij_*, were determined using density-functional perturbation theory (DFPT). Consequently, the longitudinal piezoelectric coefficients, *d*_33_, could be estimated using *d*_33_ = *e*_33_/*C*_33_.

## 3. Results and Discussions

### 3.1. Morphology and Structural Property

To fabricate the ultrathin Bi_2_Se_3_ nanosheets, mechanical exfoliation from bulk Bi_2_Se_3_ using scotch tape was performed on an n-doping Si substrate with 300 nm-thick thermal oxide layers. Raman spectra of the nanosheets were obtained using a 632.8 nm laser, and the results were presented in Figure 1a. The position and shape of the nanosheets were observed using an optical microscope (OM), revealing rectangular shapes with domain sizes of several micrometers and colors resembling the substrate, as shown in the inset of Figure 1a. The characteristic peaks at 133 and ~176 cm^−1^ correspond to the $E_{g}^{2}$ and $A_{1g}^{2}$ vibrational modes, respectively, consistent with the prior research [27]. Furthermore, atomic force microscopy (AFM) was employed to determine the thickness of the Bi_2_Se_3_ nanosheet shown in Figure 1b. The rectangular Bi_2_Se_3_ nanosheet was found to be approximately 5 μm long and 2 μm wide. The inset of Figure 1b depicts the height profile along the green line, demonstrating a thickness of approximately 5 nm. We also observed the ultrathin Bi_2_Se_3_ nanosheets using a transmission electron microscope (TEM). Figure 1c displays the TEM image of the Bi_2_Se_3_ flake. Additionally, the corresponding Selected Area Electron Diffraction (SAED) spot pattern is shown in Figure 1c to confirm the single-crystalline quality of the nanosheet. The HRTEM image of the corresponding area is displayed in Figure 1d. A lattice fringe is visible, with a lattice space of 0.31 nm, consistent with the (015) plane of the rhombohedral phase of Bi_2_Se_3_ (Figure 1c). 

### 3.2. SPM Characterization

Considering the limitations of the DART mode, we have implemented an advanced approach based on the high-throughput sequential excitation combined with piezoresponse force microscopy (SE-PFM) [20,21,22]. As depicted in Figure 2a for a schematic representation, the SE model requires only a single scan to collect the response of the sample. The key technique of the SE-PFM involves applying a high-resolution wave to the Bi_2_Se_3_ sample in the time domain, with the frequency increasing over time. This drive signal is formed by *m* jointed sinusoidal waves with discrete frequencies (depicted on the left of Figure 2a). Each pixel during the scan can generate this signal using an Arbitrary Waveform Generator (AWG). In the time domain, a feedback signal is obtained, while in the frequency domain, a signal is acquired through Fourier transform (shown on the right side of Figure 2a). As a result, for each pixel point, m pairs of amplitude $\;A\left(\omega_{i}\right)$ and phase $\emptyset \left(\omega_{i}\right)\;$ are obtained. To reduce background noise, the principle component analysis (PCA) [20,21,22] is used, and the obtained datasets are employed to compute intrinsic parameters such as amplitude, phase, quality factor, and resonant frequency using the SHO model. As a result, the magnitude of the piezoelectric response for piezoelectric samples is represented by amplitude (Figure 2b), while the orientation of polarization is depicted by the phase (Figure 2c) using SE-PFM. The results demonstrate a strong response both in amplitude and phase mapping, distinct from the substrate, suggesting the presence of piezoelectricity. It is important to note that the resonant frequency and quality factors are associated with the contact stiffness and energy dissipation of the system [24]. For comparison, the DART mode was utilized to characterize the out-of-plane piezoresponse of the same sample (Figure 2d,e). We compared the intrinsic amplitude and resonant frequency mapping of the out-of-plane piezoresponse obtained from both SE and DART modes. Both methods exhibit a strong response in amplitude and phase mapping, different from the substrate. However, there is a slight difference between the two modes; the SE mode, compared to the DART mode, provides a better visualization of the piezoelectric response region. This is because the DART mode only measures amplitude and phase at two frequencies, which may lack accuracy and reliable resonance frequency. 

It is thus intriguing to investigate the possibility of reversing the polarity of the Bi_2_Se_3_ nanosheet. To further examine its polarization switching state, we conducted switching spectroscopy by piezoresponse force microscopy (SS-PFM). In order to minimize the influence of electrostatic force, the piezoelectricity performance was measured in the “off-state”. Figure 2f depicts the switchable hysteretic behavior that aligns with the applied bias V_dc_ (10 V) in SS-PFM. The amplitude response presents an expected butterfly shaped hysteresis loop. At a relatively high voltage, the amplitude saturates, indicating a piezoelectric response rather than an electrostatic one. Figure 2f also illustrates the hysteresis behavior, displaying a sharp switching of electrical polarization with 180° applied DC bias. 

To further validate the linear piezoelectric behavior in the as-obtained Bi_2_Se_3_ nanosheet, we performed point-wise first and second harmonic measurements [22,28,29] in vertical modes. Following the approach demonstrated by Chen et al. [30], comparing first and second-harmonic PFM responses, we can distinguish the underlying mechanism. Nonlinear mechanisms, unlike piezoelectricity, often result in a prominent second harmonic response. The first harmonic test exhibits a linear response, indicating tip deflection and excitation and measurement at the fundamental contact resonance of $\;f_{0}$. On the other hand, the second harmonic test reveals a secondary response of tip deflection, excited at $f_{0}/2\;$ but measured at $f_{0}$ [28]. This measurement was carried out with an excitation bias range of ~2.3 V, and six spatial points were sampled across the Bi_2_Se_3_ nanosheets and the SiO_2_ substrate. Figure 2g shows that the first harmonic response shows linearity with respect to the applied bias and dominates over the second harmonic quadratic response. This observation indicates that the electromechanical response from Bi_2_Se_3_ nanosheets is attributed to linear piezoelectricity. In addition, we extracted the first and second harmonic response of the sample at a driving voltage of 2 V in point-wise statistics, and the highest peaks were observed at 330 kHz (inset of Figure 2g). Therefore, it is evident that the first harmonic responses significantly surpass the second harmonic responses, confirming the presence of linear piezoelectricity in atomically thin Bi_2_Se_3_ nanosheets. In summary, our findings confirm that the out-of-plane electromechanical response in Bi_2_Se_3_ originates from linear piezoelectricity and not from ionic electrochemical dipoles induced by the charged probe.

### 3.3. Relaxation Dynamics

The relaxation phenomenon, measured using ESM, is influenced by both the concentration and diffusivity of respective ionic types [31]. Hence, we employ relaxation studies to investigate the local ionic dynamics on the surface of the Bi_2_Se_3_ nanosheet. In this approach, a direct-current (DC) voltage is applied on top of an alternating current (AC) to induce the movement of surface ions underneath the probe, driven by the topological surface state, as illustrated schematically in Figure 3a. After the DC voltage is removed, the surface ions return to their initial state, and the local dynamics can then be deduced from the time constant associated with the relaxation of the ESM amplitude. Firstly, the ESM amplitude increases under negative and positive DC voltages, with a more prominent increment observed under positive DC. The ESM amplitude drops and relaxes to equilibrium after removing either the negative or positive DC voltage. These observations suggest that the local electrochemical strain, as measured by ESM, predominantly arises from polarized charges rather than the Vegard strain. Notably, after applying and removing a positive DC voltage, the amplitude is lower than the initial state, while the same conditions applied to a negative voltage result in a higher amplitude.

Interestingly, after removing both negative and positive DC voltages, the ESM amplitude consistently decreases, as indicated by the red and orange boxes at the bottom of Figure 3a. This phenomenon is further highlighted in the zoomed-in relaxation curves shown in Figure 3b,c, where the buffer times for the downtrend of the ESM amplitude are 0.7 s and 0.2 s after removing positive and negative DC voltages, respectively. Secondly, the phases of the ESM response under negative and positive DC voltages are opposite. The phase under positive DC is the same as that in the initial state, while the phase after the removal of positive DC jumps down by approximately 180°, resembling the situation under the negative DC voltage. Understanding the corresponding mechanisms related to the topological state is of significant importance. A schematic diagram illustrating the piezoelectric response is presented in Figure 3d: when a voltage is applied to the sample, the electron cloud around the atom undergoes changes. After removing the voltage, the electron cloud requires time to return to its original state. This may be the source of the relaxation phenomenon.

### 3.4. Density Functional Theory (DFT) Calculations

To validate our experimental findings, we conducted DFT calculations on Bi_2_Se_3_ in bulk and layered phases (Figure 4a) to examine their piezoelectric properties. In the Bi_2_Se_3_ slab, layers ranging from one to five were considered. The calculated *d*_33_ values for both Bi_2_Se_3_ bulk and 1 QL (monolayer) were zero, indicating no piezoelectric response. However, for Bi_2_Se_3_ slabs with 2 QL to 5 QL, *d*_33_ increases monotonically from 3.01 to 156.11 pm·V^−1^, as shown in Figure 4b, indicating the presence of a piezoelectric effect in these systems. Piezoelectric materials must satisfy two fundamental conditions: insulating and breaking central inversion symmetry. Previous reports have confirmed that the Bi_2_Se_3_ bulk and slabs from 1 QL to 5 QL possess a band gap, ref. [32] implying the insulating nature of these structures. Hence, the symmetry of the structure plays a crucial role in determining their piezoelectric properties. Due to its central inversion symmetry in the R-3m (No. 166) space group of the Bi_2_Se_3_ bulk phase, no piezoelectric effect is expected. In the 1 QL slab, the space group changes to P-3m1 (No. 164), but the inversion symmetry center is still present, resulting in the absence of a piezoelectric effect. However, as the number of Bi_2_Se_3_ layers increases from 2 QL to 5 QL, the space group (P3m1, No. 156) differs from that of the 1 QL slab, leading to the disappearance of the inversion symmetry center. This observation supports our PFM measurements. Importantly, it should be emphasized that 5 QL Bi_2_Se_3_ exhibits a significant piezoelectric effect in the out-of-the-plane direction with a *d*_33_ value of 156.11 pm·V^−1^. This piezoelectric magnitude is comparable to the best-performing inorganic ferroelectrics without alloying, making it a promising candidate for piezoelectric devices.

## 4. Concluding Remarks

In this study, we have comprehensively characterized the piezoelectric properties of the topological insulator at the surface of the odd-layer Bi_2_Se_3_ nanosheets using the scanning probe microscopy technique. The results confirm a solid piezoelectric response in this material. To further investigate the piezoelectricity of Bi_2_Se_3_, we employed both conventional PFM and the SE-SPM technique, which provided enhanced spatial resolution and high quantitative fidelity. Furthermore, point-wise measurements of the first and second harmonic responses demonstrated that the electromechanical response in Bi_2_Se_3_ arises from linear piezoelectricity rather than ionic electrochemical dipoles induced by the charged probe. Moreover, we observed that the polarity of Bi_2_Se_3_ can be switched, as evidenced by the characteristic hysteresis and butterfly loops. An intriguing finding in our study is a relaxation phenomenon on the surface of Bi_2_Se_3_ nanosheets. When the Bi_2_Se_3_ nanosheet is reduced to an odd thin layer, surface termination within a single quintuple layer or surface doping leads to an asymmetric structure and non-intrinsic piezoelectricity. We believe that the discovery of piezoelectricity in odd layers of Bi_2_Se_3_ nanosheets lays a solid foundation for further research in this field.

## Figures and Tables

**Figure 1 nanomaterials-13-02504-f001:**
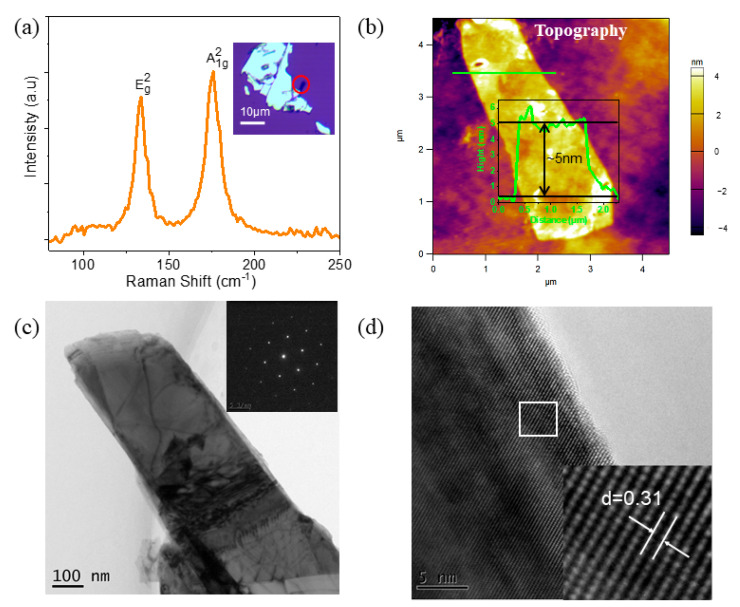
(**a**) Raman spectroscopy using a 633 nm laser on the nanosheet. The inset is an optical microscopic image of Bi_2_Se_3_ nanosheet on Si/SiO_2_ substrate by mechanical exfoliation. (**b**) Corresponding AFM images and height profiles along the green line (the inset image). (**c**) TEM characterizations of exfoliated Bi_2_Se_3_ nanosheets; the inset is the SAED pattern acquired from the same Bi_2_Se_3_ nanosheets. (**d**) HRTEM image of one typical Bi_2_Se_3_ nanosheet.

**Figure 2 nanomaterials-13-02504-f002:**
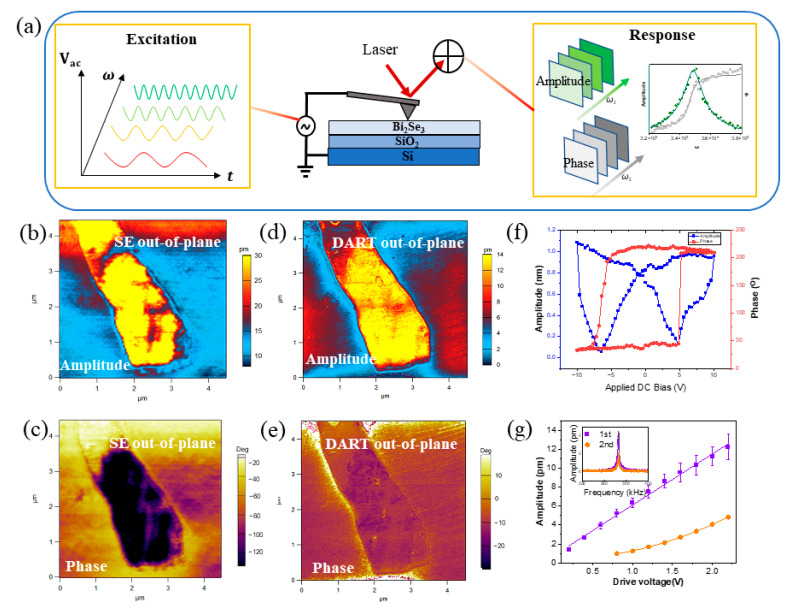
PFM characterization of Bi_2_Se_3_ on Si/SiO_2_ substrate. (**a**) Schematics of SE-PFM. (**b**) Amplitude and (**c**) phase mapping of the sample by SE mode in out-of-plane direction. (**d**) Amplitude and (**e**) phase mapping by DART mode. (**f**) Amplitude and phase by SS-PFM in “Off-state”. (**g**) Point-wise linear and quadratic VPFM responses and switching characteristics; inset is the single-point (2 V) linear and quadratic VPFM responses.

**Figure 3 nanomaterials-13-02504-f003:**
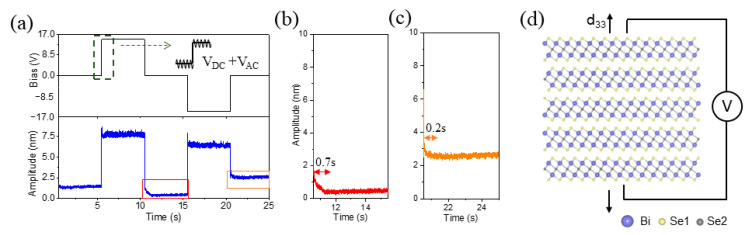
Relaxation dynamics of local electrochemical strain in Bi_2_Se_3_ nanosheet at room temperature. (**a**) Schematic DC profile for relaxation measurement (top) and ESM amplitude (middle), and phase (bottom) versus time corresponding to the DC profile. (**b**,**c**) Zoomed-in relaxation curves after removing negative and positive DC. (**d**) Schematic diagram of piezoelectric response.

**Figure 4 nanomaterials-13-02504-f004:**
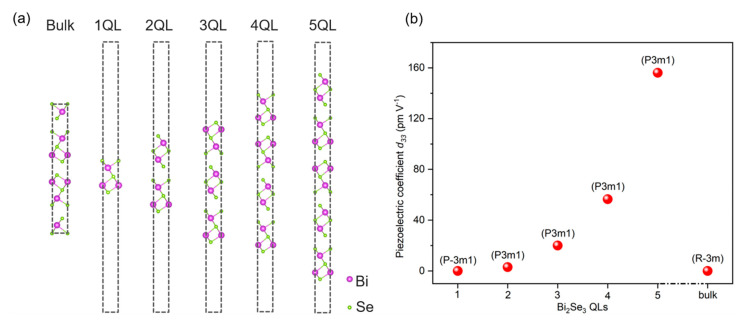
(**a**) The optimized structure of Bi_2_Se_3_ bulk and layers from 1 QL to 5 QL and the (**b**) corresponding piezoelectric coefficient *d*_33_. The space group of these structures is marked as well.

## Data Availability

The data that support the findings of this study are available from the corresponding author upon reasonable request.

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
