# Peer review of "Piezoelectricity of Bi2Se3 Nanosheet"

_nanomaterials, 2023, doi:10.3390/nano13182504_

Round 1
Reviewer 1 Report
The work «Piezoelectricity of Bi2Se3 Nanosheet» is devoted actual and significant topic about the piezoelectric response in a 5-layer Bi2Se3 nanosheet using scanning probe microscopy(SPM) techniques. The abstract briefly describes the motivation and results of the study. In the introduction, the authors describe well the problem being solved and the proposed ways for this. All synthesis methods contain a detailed description of the operations performed. The results obtained are presented in the form of figures of good quality and informative. The revealed regularities have an explanation. And I have some comments:
1. One or two sentences are required at the beginning of abstract to present the background or aim of this work.
2. I recommend that the authors check the necessity of all equations presented in the manuscript. In my opinion, equation 1 is not used in any way, if I'm wrong, then add information.
3. There are too many too old references, which is better to be deleted or replaced with recent articles to show the novelty of this work.
Author Response
Comment 1.
One or two sentences are required at the beginning of abstract to present the background or aim of this work.
Response
Thanks for the advice. We have added the background description in the revision.
Comment 2.
I recommend that the authors check the necessity of all equations presented in the manuscript. In my opinion, equation 1 is not used in any way, if I'm wrong, then add information.
Response
Equation 1 disctribes the interactions between sample and tip, which is the basic principle of PFM technique. The new technique SE-PFM described in this work is also based on the equation. We have described in the manuscript. Please check the paragraph in line 54 to70 on page 2.
Comment 3.
There are too many too old references, which is better to be deleted or replaced with recent articles to show the novelty of this work.
Response
Thanks for the advice. We add new references on the latest progress in the revision.

Reviewer 2 Report
In this manuscript, piezoelectric properties of 5-layer BiSe3 nanosheets have been reported. The nanosheets of BiSe3 were obtained by following mechanical exfoliation method upon Si/SiO2 substrate. Extensive piezoelectric characterization involving Dual amplitude resonance tracking (DART)-PFM, sequential excitation (SE)-PFM and point-wise measurement of the first and second harmonic responses suggests linear intrinsic piezoelectricity in the nanosheets. DFT calculation shows the center of inversion in the crystal structure for 2-to-5-layer Bi2Se3 nanosheets and supports the experimentally observed piezoelectric nature of the nanosheets. The manuscript is well-written and can be accepted with some minor corrections. The comments are below:
1. Please improve the English language and keep the font size uniform throughout the manuscript.
2. Include years in the Refs. 12, 16, and 20 in the reference section.
3. Include the page number for Refs. 14, 15, 21, 24 and 26.
4. Please include the recent citation on studies of piezoelectricity of similar chalcogenides in nanosheets form in the introduction section on P. 1-2.
Moderate editing of English language required
Author Response
Thanks for pointing out the mistakes in our manuscript. We have carefully revised the paper and made the corrections in the revision.